# An Integrated Analytical Approach Reveals Trichome Acylsugar Metabolite Diversity in the Wild Tomato *Solanum pennellii*

**DOI:** 10.3390/metabo10100401

**Published:** 2020-10-09

**Authors:** Daniel B. Lybrand, Thilani M. Anthony, A. Daniel Jones, Robert L. Last

**Affiliations:** 1Department of Biochemistry and Molecular Biology, Michigan State University, East Lansing, MI 48824, USA; dblybrand@ucdavis.edu (D.B.L.); thilani@chemistry.msu.edu (T.M.A.); jonesar4@msu.edu (A.D.J.); 2Department of Plant Biology, Michigan State University, East Lansing, MI 48824, USA

**Keywords:** acylsugar, wild tomato, *Solanum pennellii*, secretory glandular trichome, specialized metabolism, intraspecific variation, metabolomics

## Abstract

Acylsugars constitute an abundant class of pest- and pathogen-protective Solanaceae family plant-specialized metabolites produced in secretory glandular trichomes. *Solanum pennellii* produces copious triacylated sucrose and glucose esters, and the core biosynthetic pathway producing these compounds was previously characterized. We performed untargeted metabolomic analysis of *S. pennellii* surface metabolites from accessions spanning the species range, which indicated geographic trends in the acylsugar profile and revealed two compound classes previously undescribed from this species, tetraacylglucoses and flavonoid aglycones. A combination of ultrahigh-performance liquid chromatography–high resolution mass spectrometry (UHPLC–HR-MS) and NMR spectroscopy identified variations in the number, length, and branching pattern of acyl chains, and the proportion of sugar cores in acylsugars among accessions. The new dimensions of acylsugar variation revealed by this analysis further indicate variation in the biosynthetic and degradative pathways responsible for acylsugar accumulation. These findings provide a starting point for deeper investigation of acylsugar biosynthesis, an understanding of which can be exploited through crop breeding or metabolic engineering strategies to improve the endogenous defenses of crop plants.

## 1. Introduction

Plants produce thousands of lineage-specific compounds, termed specialized metabolites [1]. Unlike the highly conserved core pathways common to nearly all plants, specialized metabolic pathways evolve rapidly, leading to tremendous structural and functional diversity (e.g., terpenoids and flavonoids) [2,3,4]. These pathways and their products provide a chemical palette that mediates interactions between plants and biotic or abiotic stressors in their environments. Many of these specialized metabolites accumulate in specialized structures, including epidermal secretory glandular trichomes (SGTs) [5], which act as a first line of defense against herbivores and pathogens [6,7,8,9].

Plants of the genus *Solanum*, which include tomato, potato, and eggplant, possess multiple types of SGTs that produce a diverse array of specialized metabolites [10,11], including acylsugars, which accumulate up to 20% of leaf dry mass in the wild tomato *Solanum pennellii* [12]. *S. pennellii* acylsugars consist of sucrose or glucose cores esterified with fatty acid acyl groups of variable length and branching pattern (Figure 1). These compounds defend *S. pennellii* and other *Solanum* species from insect pests including silverleaf whitefly (*Bemisia tabaci*), western flower thrips (*Frankliniella occidentalis*), and army beetworm (*Spodoptera exigua*) [13,14,15]. The antioviposition and antiherbivory properties of *S. pennellii* acylsugars prompted efforts to breed cultivated tomato (*Solanum lycopersicum*) varieties with *S. pennellii*-like acylsugar profiles [13,15,16]. Such efforts are aided by knowledge of the genetic loci underlying acylsugar biosynthesis [15,16,17,18]. The *S. pennellii* core acylsugar biosynthetic pathway consists of three BAHD(Benzyl alcohol *O*-acetyltransferase, anthocyanin *O*-hydroxycinnamoyltransferase, *N*-hydroxycinnamoyl/benzoyltransferase, deacetylvindoline 4-*O*-acetyltransferase)-family acylsugar acyltransferases (ASATs) that sequentially transfer acyl groups from coenzyme A (CoA) donors to a sucrose acceptor, yielding triacylsucroses (Figure 1) [19]. As shown in Figure 1, enzymes involved in acyl CoA biosynthesis (e.g., acyl CoA synthetase (ACS), enoyl CoA hydratase (ECH), and isopropylmalate synthase 3 (IPMS3)) affect the structures of triacylsucroses by modulating the available pool of acyl CoA donors [20,21], while enzymes that selectively cleave acyl chains from intact acylsugars (acylsugar acylhydrolases (ASHs)) or hydrolyze triacylsucroses to form triacylglucoses (acylsucrose fructofuranosidase1 (ASFF1)) influence steady-state acylsugar profiles and facilitate rapid acylsugar turnover [22,23,24,25].

Previous studies of acylsugar metabolism in *S. pennellii* focused on either the chemical substructures of acylsugars (i.e., sugar cores and acyl chains; [21,26,27]) or the enzymes that synthesize and degrade acylsugars (ACS, ASATs, ASFF1, ASHs, ECH, IPMS3) [19,20,21,22,23]. This work revealed that *S. pennellii* accumulates a mixture of acylglucoses and acylsucroses [26,28], collectively containing acyl chains with at least 13 unique structures [21,26,27]. Although most studies focused on the southern Peruvian *S. pennellii* LA0716 [29] acylsugars, [19,22,23,27,30,31,32,33], Shapiro and co-workers quantified abundance of acylsugar substructures from 19 accessions of *S. pennellii* distributed across the range of the species [26], while Ning and co-workers analyzed acylsugar acyl chains in 14 *S. pennellii* accessions to determine the genetic basis for differential accumulation of 3-methylbutanoate and 2-methylpropanoate acyl chains in northern and southern regions of Peru [21]. Knowledge of relative acylsugar substructure abundances among related species or populations provides an insight into the biosynthesis of these compounds, which facilitates the use of acylsugars in crop defense and illuminates the evolution of specialized metabolic pathways [15,21].

A complete understanding of acylsugar biosynthesis and evolution requires a knowledge of specific acylsugar structures, as revealed by metabolomic approaches, including untargeted liquid chromatography–mass spectrometry (LC–MS) of intact molecules or structural resolution by NMR spectroscopy. For example, an early report on *S. pennellii* acylsugar metabolism in which Burke and co-workers partially characterized acylglucoses in *S. pennellii* LA0716 by NMR [28] provided information on the number of acyl chains and established esterification on the two-, three-, and four-positions, which later facilitated discovery of the three ASATs constituting the core acylsugar biosynthetic pathway [19,34]. A combination of untargeted metabolomic analysis of acylsugars in *Solanum habrochaites* and *Petunia axillaris* [35,36] using LC–MS and NMR spectroscopy facilitated elucidation of core acylsugar pathways in these species [19,37].

Structural information about in planta intact acylsugars is essential for discovering and characterizing enzymes in the acylsugar biosynthetic pathway of a single species and for comparing pathways between species. To create a more complete picture of acylsugar diversity in *S. pennellii*, we combined untargeted ultrahigh-performance liquid chromatography–high resolution mass spectrometry (UHPLC–HR-MS) and NMR spectroscopy to characterize the SGT metabolome of 16 *S. pennellii* accessions, revealing variations in levels of 43 specialized metabolites, including 39 acylsugars. We initially annotated all metabolites based on mass spectra and subsequently purified and resolved structures of selected acylsugars by NMR. Multivariate statistical analyses of these profiling data recognized specific compounds that distinguish various *S. pennellii* accessions from one another. Our analyses confirmed previous reports showing that acyl chain complement drives acylsugar variation between *S. pennellii* accessions [21,26], and revealed a positive correlation between expression of the *ASFF1* gene that facilitates acylsucrose hydrolysis [22] and acylglucose accumulation. We also observed tetraacylglucoses and methyl flavonoids, two classes of compounds previously undescribed in *S. pennellii* SGTs.

## 2. Results

### 2.1. Experimental Design

Previous studies indicated that wild tomato species, including *Solanum pennellii*, exhibit intraspecific variation in the amounts and types of acylsugars produced [21,26,38]. To identify geographic trends in acylsugar quantity and quality in *S. pennellii*, we extracted compounds from the surfaces of leaflets from 16 accessions spanning the 1500-km geographic range of the species (Figure 2). Six biological replicates of each accession were sampled to capture intra-accession metabolic diversity. We included eight accessions from the northern portion of Peru (north range) and eight from the southern portion (south range) and classified two clusters of accessions within the south range by region, including the southernmost Atico group and the Pisco group. A group of accessions from the Nazca region, described as *S. pennellii var. puberulum*, are trichome-deficient and exhibit the minimal accumulation of acylsugars and transcripts of genes associated with acylsugar metabolism [26,39]; our pilot experiments confirmed the absence of detectable acylsugars in this group and these accessions were excluded from this study. All extracts were analyzed by UHPLC–HR-MS using positive-mode electrospray ionization.

### 2.2. Untargeted Metabolomics Reveals Acylsugars and Flavonoids in Trichomes

Automated feature extraction and deconvolution of compound ions detected by UHPLC–HR-MS analysis of leaf dip extracts followed by filtering to remove low-quality features resulted in the detection of 54 metabolic features. Based on the annotation of collision-induced dissociation (CID) spectra and comparisons to previously characterized trichome-localized metabolites in *Solanum* spp. [23,28,40], we categorized all 54 metabolic features as putative acylsugars or flavonoid aglycones. All annotated acylsugars possessed either a six-carbon monosaccharide core or a 12-carbon disaccharide core based on an analysis of neutral losses from pseudomolecular ions and *m/z* from product ions.

Acylglucoses sharing acylation patterns are resolved as distinct α and β anomers by reverse-phase chromatography, but some acylglucose β anomers co-elute with α anomers of later-eluting acylglucose isomers, precluding the direct determination of the number of acylglucoses present in a sample from the number of acylsugar metabolic features detected. An examination of chromatograms and associated spectra for all features categorized as acylglucoses revealed that 32 metabolic features with distinct retention times identified as acylglucoses collectively represent α and β anomers of 21 acylglucoses containing unique acyl chain complements. This consolidation reduced the 54 metabolic features assigned in silico to 43 metabolic features. We categorized these features as 18 triacylsucroses, 19 triacylglucoses, two tetraacylglucoses, and four flavonoids (Table 1 and Table 2).

### 2.3. Acylsugar Core Composition Varies across the S. pennellii Geographic Range

The absolute quantification of total acylsucroses and acylglucoses in 16 accessions of *S. pennellii* revealed the variation in total acylsugar accumulation (from 133 µmol/g dry weight (DW) in LA2657 to 340 µmol/g DW in LA2560) and the relative abundance of acylglucoses and acylsucroses (from 42% acylglucoses in LA2963 to 95% acylglucoses in LA0716) (Table 3; Appendix A). While we found no discernable geographic trends in total acylsugar accumulation (Table 3; Appendix A), a higher relative abundance of acylglucoses was observed in southern accessions compared with northern accessions. In the northern extent of the range, acylglucose composition varied from 56% (LA2657) to 70% (LA2719), while in the southern span, values ranged from 77% (LA1693) to 95% (LA0716) acylglucose. The south accession LA2963 is a notable exception to this trend, showing a lower acylglucose composition (42%) than any other accession (Table 3; Appendix A).

### 2.4. Variable Acyl Chain and Sugar Composition Yield Acylsugar Diversity

The annotation of the 39 acylsugars present in our dataset revealed 26 unique molecular formulas, including multiple structural isomers (Table 1). Initial annotations were based on exact pseudomolecular and fragment masses with comparisons to homologous metabolites previously described. We had confidence in our annotations using the Metabolomics Standards Initiative guidelines [41], and they are presented in Table 1 and Table 2. Compounds whose structures were established by NMR are designated with a confidence level of one. As no alternative sugar cores other than glucose and sucrose have been reported from *S. pennellii*, this isomerism is likely driven by variations in acyl chains or positions of specific acylations. Six pairs of structural isomers had indistinguishable high-energy CID mass spectra (Figure 3A; Table 1). This suggests two possible non-mutually exclusive types of acylsugar structural isomerisms: acylsugars with similar complements of acyl chains, but differing in acyl chain positions (positional isomers), and acylsugars bearing acyl chains with identical chemical formulas, but different branching patterns (acyl chain structural isomers). The latter hypothesis is supported by previous reports of unbranched, iso-branched, and anteiso-branched acyl chains in *S. pennellii* acylsugars [18,21,26]. Additional structural isomers differ in the number of carbons present in individual constituent acyl chains (Figure 3B; Table 1), while keeping the total number of acyl carbons constant. The presence of two tetraacylglucoses in the dataset, G4:14(2,4,4,4) and G4:15(2,4,4,5), also indicates variation in the number of acylsugar acylations. In contrast, neither mono- nor di-acylated sugars were observed, although these intermediates were reported from in vitro studies of triacylsucrose biosynthesis and in planta in lines bearing introgressions of *S. pennellii* in the *S. lycopersicum* M82 background [19,42].

All but one of the annotated triacylsucroses [S3:17(4,4,9)] in our dataset show patterns of acyl group neutral losses in their mass spectra that mirror those observed in at least one triacylglucose (Table 1). We hypothesized that pairs of acylsucroses and acylglucoses with similar neutral mass losses possessed identical acyl chain complements, consistent with the current model of *S. pennellii* acylsugar biosynthesis in which cleavage of acylsucrose glycosidic bonds by ASFF1 removes the β-fructofuranose rings to yield acylglucoses [22]. These observations indicate that variations in the identity of acyl chains, the number of acyl chains, and identity of sugar core all contribute to the acylsugar diversity in *S. pennellii*. While the presence of multiple acylsugar structural isomers with identical mass spectra implies isomeric acyl chains, and the similarity in neutral losses between acylsucroses and acylglucoses suggests identical chain elemental composition but not necessarily topology; the mass spectrometry techniques applied could not establish key structural features, leading us to resolve their structures using NMR.

### 2.5. NMR Spectroscopy Resolves Structural Relationships between Acylsugars

We selected 10 acylsugars for purification and structural resolution by NMR, including five triacylsucroses (S3:12(4,4,4), S3:18(4,4,10)-1, S3:18(4,4,10)-2, S3:19(4,5,10)-1, and S3:19(4,5,10)-2) and five triacylglucoses (G3:12(4,4,4), G3:18(4,4,10)-1, G3:18(4,4,10)-2, G3:19(4,5,10)-1, and G3:19(4,5,10)-2) (Table 1; Figure 4; See Appendix A, Appendix A for NMR chemical shifts and spectra). These compounds were selected due to the high abundance of the specified acylglucoses in *S. pennellii* LA0716 and structurally similar acylsucroses in *S. pennellii* LA0716 *asff1-1* mutants. NMR spectroscopy confirmed that all examined disaccharide-containing acylsugars possess a sucrose core while all monosaccharide acylsugars are based on glucose, consistent with previous analyses of *S. pennellii* acylsugars [23,26,28]. NMR analysis further revealed that all are acylated at the 2-, 3-, and 4- hydroxyls of the pyranose ring, also consistent with previous reports [23,28]. The structures of two compounds, G3:12(4,4,4) and S3:19(4,5,10)-1, matched two previously published acylsugar structures [23,28].

We tested the hypotheses that acylsugar isomers with indistinguishable mass spectra possess either identical complements of acyl chains attached to different positions of the sugar core or isomeric acyl chains with different branching patterns. The structures of four pairs of isomers were compared, including two pairs each of acylsucrose and acylglucose isomers (S3:18(4,4,10)-1/2, S3:19(4,5,10)-1/2, G3:18(4,4,10)-1/2, G3:19(4,5,10)-1/2). In each case, both isomers had identical configurations of acyl chains at the 2- and 4- positions. However, for all four isomeric pairs, we observed an iso-branched 10-carbon acyl chain (R3 = (Me)_2_CH(CH_2_)_6_) in the earlier-eluting isomer and an unbranched 10-carbon acyl chain (R3 = Me(CH_2_)_8_) in the later-eluting isomer (Figure 4). This demonstrates that acylsugar diversity is influenced by differences in acyl chain branching patterns as well as variation in the molecular formulas of constituent acyl chains. We also compared the structures of acylsucroses and acylglucoses with similar neutral loss patterns. The acylation pattern of each of the five purified acylsucroses was identical to that of their analogous purified acylglucoses (e.g., S3:12(4,4,4) and G3:12(4,4,4); S3:19(4,5,10)-1 and G3:19(4,5,10)-1; Figure 4). This is consistent with the hypothesis that these acylsucroses are biosynthetic precursors of the analogous acylglucoses.

### 2.6. Flavonoids Vary by Core and Degree of Methylation

Our dataset contained four methylated flavonoid aglycones. Methyl flavonoid molecular formulas were consistent with di-, tri-, and tetramethylated derivatives of tetra- and pentahydroxylated flavonols (Table 2), resembling the methylated myricetins observed in *S. habrochaites* and *S. lycopersicum* [40,43,44]. Two kaempferol-like tetrahydroxylated flavonoids were observed possessing two and three methylations (denoted as flavonoids A and B), while two quercetin-like pentahydroxylated flavonoids were observed, possessing three and four methylations (flavonoids C and D). As *S. lycopersicum* accumulates glycosylated derivatives of the flavonols kaempferol and quercetin (tetra- and pentahydroxylated, respectively) in type VI trichomes [6,44], we hypothesized that the methylated flavonoids observed in *S. pennellii* leaf dips were also kaempferol- and quercetin-derived. While an analysis of the flavonoid mass spectra indicated molecular formulas and the presence of methyl groups, few low-mass fragment ions were present in the spectra to aid in further structural assignment, previously demonstrated with myricetin derivatives (Appendix A) [45]. Nevertheless, our results indicated flavonoid diversity in terms of both flavonol core and degree of methylation.

### 2.7. Multivariate Analysis Implicates Short-Branched Acyl Chains in North-South Acylsugar Variation

We used the full dataset representing 43 specialized metabolites in 16 accessions of *S. pennellii* to identify metabolite-based differences between accessions across the geographic range. Due to overlapping retention times observed with some acylglucose anomers and the resulting difficulty in assigning accurate abundances to individual acylglucoses, we used the original dataset containing 54 metabolite features obtained prior to spectral interpretation instead of the dataset containing the 43 unique metabolites. An unsupervised principal component analysis (PCA) of metabolites’ signal abundances of all accessions revealed clear separation of accessions in the north range from those in the south range with the exception of two outliers (Figure 5). These samples both represent individuals of south range accession LA1946 that cluster with north range samples; we hypothesize that this is due to seed contamination or sample tracking errors. 

An orthogonal partial least squares/projection to latent structures discriminant analysis (OPLS–DA) model successfully classified 100% of north range samples and 94% of south range samples (Table 4), indicating that metabolite features ranked by the model were good predictors of geographic origin. Metabolite features were ranked using the correlation values obtained from the OPLS–DA model (Appendix A; Appendix A). Negative correlation values indicate correlation with north region accessions while positive correlation values indicate correlation with south region accessions. Structural characteristics of the five metabolite features showing the strongest quantitative correlation with either sample class were compared. Three acylglucoses [G3:15(5,5,5), G3:16(5,5,6), G3:21(5,5,11)] and two acylsucroses [S3:16(5,5,6), S3:21(5,5,11)] showed the strongest correlation with north range accessions, while four acylglucoses (G3:12(4,4,4), G3:13(4,4,5), G3:18(4,4,10)-2, G3:19(4,5,10)-2) and one acylsucrose (S3:18(4,4,10)-2), showed the strongest correlation with south range accessions. Four acylsugars enriched in the south range (G3:12(4,4,4), G3:18(4,4,10)-2, G3:19(4,5,10)-2, S3:18(4,4,10)-2) were structurally characterized by NMR in this study (Figure 4), while a fifth [G3:14(4,5,5)] was annotated in a previous work [23]. All four-carbon acyl chains in these acylsugars are 2-methylpropanaote, while only one of the five-carbon chains in the G3:14(4,5,5) compound is 3-methylbutanoate; the other five-carbon acyl chains in G3:14(4,5,5) and G3:19(4,5,10)-2 are 2-methylbutanoate. While we cannot definitively identify the branching pattern of five-carbon acyl chains in the metabolites associated with the north range, our findings agree with previously observed trends in *S. pennellii* favoring accumulation of four-carbon 2-methylpropanoate chains in southern accessions and five-carbon 3-methylbutanoate chains in northern accessions, with five-carbon 2-methylbutanoate chains abundant across the range [21,26].

### 2.8. Variation in Medium-Length Acyl Chains Drives Variation within the South Range

As our PCA also indicated substantial intragroup variation in south range samples (Figure 5), we performed additional multivariate analyses to distinguish profiles within south range plant extracts (Figure 6). These accessions form two distinct geographic clusters from the Pisco or Atico regions (Figure 2). An OPLS–DA model of Pisco and Atico samples successfully classified 67% of Pisco region samples and 77% of Atico region samples but misclassified or was unable to classify 28% of all samples (Table 4), indicating that this model performed poorly when compared to our north/south range OPLS–DA model. However, the model still recognized metabolites that had a strong quantitative correlation with either the Pisco or Atico region samples. Metabolite features were ranked using the correlation values obtained from the OPLS–DA model (Appendix A; Appendix A). Negative correlation values indicate correlation with Pisco region accessions, while positive correlation values indicate correlation with Atico region accessions. The five compounds exhibiting the strongest correlation with Pisco region samples comprised four acylglucoses [G3:16(4,4,8)-1, G3:16(4,4,8)-2, G3:17(4,5,8)-1, and G3:17(4,5,8)-2] and one acylsucrose [S3:17(4,4,9)], while the five metabolites demonstrating the strongest correlation with Atico region samples consisted of two acylglucoses [G3:20(4,4,12), G3:21(4,5,12)], two acylsucroses [S3:20(4,4,12), S3:21(4,5,12)], and one flavonoid (flavonoid A). The medium-length acyl chains (defined here as possessing more than five carbons) in correlative features show a sharp distinction between the two regions with four of five acylsugars more abundant in the Pisco region accessions bearing an eight-carbon acyl chain and all four acylsugars that are more abundant in the Atico region containing a 12-carbon acyl chain, while four- and five-carbon acyl chains have a similar distribution between the two regions. This suggests medium-length acyl chain variation as the key driver in separation of accessions from the Pisco and Atico region.

### 2.9. LA2963 Segregates from Other Atico Region Accessions Due to High Acylsucrose Content

The bimodal clustering of Atico region samples revealed by PCA (Figure 6) indicates chemical diversity among accessions that are <150 km apart, in contrast to previously reported trends in acylsugar diversity observed between accessions separated by >1000 km (Figure 5; Appendix A) [21,26]. We explored this diversity using multivariate analysis. PCA of the four Atico region accessions (LA0716, LA1941, LA1946, and LA2963) showed two major clusters (Figure 7). One cluster contained all biological replicates of accessions LA0716 and LA1941 along with four samples of LA1946 (both LA1946 samples outside this cluster represent outliers that clustered with north range accessions in our north/south PCA (Figure 5)). The other major cluster contained all samples of accession LA2963. 

An OPLS–DA model discriminating between LA2963 samples and all other Atico region accessions correctly classified 97% of samples from the main Atico cluster and 100% of LA2963 samples (Table 4), indicating that metabolite features ranked by the model were good predictors of sample group. Metabolite features were ranked using the correlation values obtained from the OPLS–DA model (Appendix A; Appendix A). Negative correlation values indicate correlation with the main Atico cluster of accessions while positive correlation values indicate correlation with accession LA2963. The five compounds exhibiting the strongest quantitative correlation with the main Atico cluster included four acylglucoses (G3:18(4,4,10)-1, G3:18(4,4,10)-2, G3:19(4,5,10)-1, G3:20(4,4,12)) and one flavonoid (flavonoid A) while the five compounds most highly correlated with the anomalous accession LA2963 were all acylsucroses (S3:18(4,4,10)-1, S3:19(4,5,10)-1, S3:19(4,5,10)-2, S3:20(4,4,12), S3:21(4,5,12)). This indicates relative acylsucrose and acylglucose abundance as the key driver of separation between LA2963 samples and other Atico region samples, and was consistent with our initial analysis of *S. pennellii* sugar core abundance, which indicated accession LA2963 as an outlier among southern accessions that possessed low acylglucose content (Table 3; Appendix A). 

Two of the acylsucroses correlated with accession LA2963 (S3:18(4,4,10)-1, S3:19(4,5,10)-1) have structures consistent with precursors of two acylglucoses correlated with the main Atico cluster (G3:18(4,4,10)-1, G3:19(4,5,10)-1, described above; Figure 4), while a third compound correlated with LA2963 (S3:20(4,4,12)) has a fragmentation pattern consistent with a possible precursor of another Atico cluster-correlated acylglucose (G3:20(4,4,12)) (Table 1). The ASFF1 enzyme hydrolyzes acylsucroses yielding acylglucoses in *S. pennellii* LA0716 [22]. We hypothesized that low ASFF1 activity in plants of accession LA2963 relative to other Atico region accessions contributed to the low accumulation of acylglucoses in this accession and corresponding high accumulation of acylsucroses. To test this hypothesis, we investigated whether there is a correlation between the relative accumulation of acylglucoses and expression of the *ASFF1* gene by saponification of acylsugar extracts and UHPLC–MS-MS sugar core quantification, and relative quantification of *ASFF1* transcript abundance by RT-qPCR in paired leaflets from three biological replicates of *S. pennellii* LA0716 and LA2963 (Figure 8). Acylglucoses constituted 94% of acylsugars in LA0716 but only 38% of acylsugars in LA2963 (Figure 8A), consistent with our previous sugar core quantification results (Table 3; Appendix A), while *ASFF1* transcripts were 2.9-fold more abundant in LA0716 than in LA2963 (Figure 8B). Linear regression analysis indicated a positive correlation between *ASFF1* transcript abundance and the percentage of acylsugars accumulating as acylglucoses (R^2^ = 0.84; Figure 8C). This correlation supports the role of the *ASFF1* gene in determining acylsugar core composition in *S. pennellii* [22] and further suggests a role for transcriptional regulation of *ASFF1* in intraspecific sugar core variation.

## 3. Discussion

To capitalize on the protective properties of acylsugars, plant breeders are creating tomato lines with altered acylsugar profiles and increased insect resistance [15,16,17,18,46,47]. This process is facilitated by knowledge of acylsugar protective properties [13] and the genetic basis for acylsugar biosynthesis and diversity [15,16,17,18]. Characterization of the acylsugars found in *S. pennellii* is essential for elucidating and evaluating the protective benefits of specific compounds and of pathways involved in acylsugar biosynthesis and degradation. Our analysis of metabolites extracted from the surface of *S. pennellii* leaflets annotated a total of 43 specialized metabolites consisting of 18 acylsucroses, 21 acylglucoses, and four flavonoids. UHPLC–MS analysis alone indicated the presence of two tetraacylglucoses (Table 1), a type of acylsugar previously unknown in *S. pennellii*, as well as four methyl flavonoids (Table 2), a class of compounds known from the related tomatoes *S. lycopersicum* and *S. habrochaites* but previously unknown in this species [6,40,43,44,45]. A combination of UHPLC–MS and NMR spectroscopy indicated both acyl chain length and branching pattern as mechanisms of acylsugar isomerism and confirmed that acylglucose structures are consistent with acylsucrose hydrolysis products (Table 1; Figure 3 and Figure 4). Multivariate analysis of our UHPLC–MS dataset provided additional confirmation of the differential accumulation of short-branched acyl chains in acylsugars from northern and southern *S. pennellii* accessions (Figure 5; Appendix A) and revealed geographic variations between smaller sub-regions within the range of the species (Figure 6; Figure 7). Accessions from the Pisco and Atico regions were distinguished by the enrichment of eight-carbon acyl chains in the former and 12-carbon acyl chains in the latter (Appendix A). Within the Atico region, the acylsugar profile of accession LA2963 differs from that of nearby accessions primarily due to low acylglucose abundance compared to other Atico region accessions (Figure 8A). The new dimensions of acylsugar variation discovered in this work demonstrate that aspects of acylsugar biosynthesis and degradation within and beyond the core pathway await characterization.

Our findings indicate that additional acyltransferase activities involved in *S. pennellii* acylsugar biosynthesis are yet to be identified. We annotated two previously unreported tetraacylglucoses, both containing acetyl groups (Table 1). While tetraacylated sugars with acetyl groups are common in *S. habrochaites* and *S. lycopersicum*, they are absent from published analyses of *S. pennellii* acylsugars [21,26,35]. Thus far, three ASATs involved in acylsugar biosynthesis, each performing a single acylation step, have been identified in *S. pennellii* [19]. The presence of tetraacylglucoses in this species requires a fourth acylation step, which could be performed by one of the previously described acyltransferases from the *S. pennellii* acylsugar pathway (i.e., ASAT1/2/3) or by an acyltransferase not previously implicated in acylsugar biosynthesis. In *S. lycopersicum*, acetylation of triacylsucroses is performed by ASAT4 [48]. The *S. pennellii* ASAT4 locus is therefore worth investigating as a candidate acylsugar acetyltransferase in this species. Explicit searches using extracted ion chromatograms of anticipated ion masses revealed no tetraacylsucroses. These compounds may accumulate at levels below the detection threshold. Alternatively, tetraacylglucoses may not be hydrolysis products of tetraacylsucroses, but rather derived via direct acylation of triacylglucoses. Further characterization of acyltransferases is necessary to determine the origins of tetraacylated sugars in *S. pennellii*.

Our analysis also revealed previously unreported intraspecific differences in acylsugar acyl chain accumulation. Prior studies confirmed eight- and 12-carbon acyl chains in *S. pennellii* acylsugars [21,26], and we identified the differential accumulation of acylsugars containing these acyl chains between accessions from the Pisco and Atico regions (Appendix A). Differences in the abundance of eight- and 12-carbon acyl chain-containing acylsugars may reflect differences in acyl CoA availability, ASAT-catalyzed incorporation of acyl chains into acylsugars, or acylsugar turnover. Interspecific variation at genetic loci encoding enoyl CoA hydratase (ECH) and acyl CoA synthetase (ACS) enzymes leads to a high proportion of 10-carbon relative to 12-carbon acyl chains in *S. pennellii* LA0716 and a high proportion of 12-carbon relative to 10-carbon acyl chains in *S. lycopersicum* M82 [20]. Differences in substrate specificity of these enzymes among *S. pennellii* accessions could lead to variation in medium-length acyl CoA pools and the subsequent incorporation of medium-length acyl chains into acylsugars. Alternatively, variation in ASAT affinity for acyl CoAs among accessions may explain the differences in acyl chain incorporation even if similar acyl CoA pools are present across accessions. Finally, the differential accumulation of eight- and 12-carbon acyl chain-containing acylsugars may reflect differences in acylsugar turnover rates between *S. pennellii* accessions. The ASH carboxylesterase enzymes facilitate acylsugar degradation in *S. lycopersicum* and *S. pennellii*, primarily by removing acyl chains from the three-position of acylsucroses and acylglucoses [23]. NMR spectra of acylsugars in *S. pennellii* consistently show medium-length R3 chains while groups R2 and R4 are exclusively short four- or five-carbon acyl chains (Figure 4) [23], suggesting that eight- and 12-carbon acyl chains could be removed by ASHs. Mass spectra indicated the presence of eight- and 12-carbon acyl chains in our dataset but the corresponding acylsugars were not selected for purification and structural resolution by NMR. As both straight and branched eight- and 12-carbon acyl chains have been observed in *S. pennellii* acylsugars [21,26,27], further structural characterization of these acylsugars by NMR is warranted for deeper exploration of their biosynthetic origins.

In addition to uncovering the variations in acylsugar acyl chains, we identified variations in sugar core proportion within the Atico region. The proportion of acylglucoses in the southern accession LA2963 (42%) was less than half of that observed for nearby accessions from the Atico region (82-95%) (Table 3; Appendix A). We tested the hypothesis that the proportion of acylsugars accumulating as acylglucoses in *S. pennellii* could be associated with levels of the ASFF1 enzyme, which hydrolyzes acylsucroses to acylglucoses in accession LA0716 [22]. A combination of UHPLC–MS-MS and RT-qPCR demonstrated that the percentage of acylsugars accumulating as acylglucoses correlated with abundance of *ASFF1* transcripts in two accessions from the Atico region, LA0716 and LA2963 (Figure 8C). Combined with the observation that knockout of the *ASFF1* gene in *S. pennellii* LA0716 abolishes acylglucose accumulation [22], our findings suggest that low levels of ASFF1 expression lead to a low proportion of acylglucoses in *S. pennellii* LA2963. Additional work is needed to dissect the mechanism leading to differences in transcript accumulation in these southern accessions.

Our current understanding of acylsugar biosynthesis in *Solanum* was achieved primarily through interspecific comparison of acylsugar phenotypes and analysis of variation in the underlying genetic loci [19,20,21,22,48]. The intraspecific variations in *S. pennellii* acylsugar phenotype reported here provide a basis for further pathway analyses. The dimensions of acylsugar variation within *S. pennellii* are potentially linked to all known components of acylsugar metabolism, including enzymes in auxiliary pathways that generate acylsugar precursors (e.g., IMPS3, ECH, ACS), activities of the core acylsugar biosynthetic pathway (i.e., ASATs), and enzymes that degrade or remodel acylsugars (e.g., ASHs and ASFF1). The presence of tetraacylglucoses indicates undiscovered core pathway acyltransferase activity in the form of new ASATs or broader substrate specificity of existing ASATs. Differential accumulation of eight-carbon and 12-carbon acyl chain-containing acylsugars among *S. pennellii* accessions may reflect variations in the biosynthesis of medium-length acyl CoA precursors to acylsugars by enzymes like ACS and ECH, variations in the ASAT affinity for medium-length acyl CoAs in the core acylsugar pathway, or variations in ASH affinity for medium-length acyl chain-containing acylsugars during acylsugar degradation and turnover. The correlation between relative acylsugar core abundance and *ASFF1* expression indicates a role for gene regulation in affecting acylsugar composition. Further investigation of acylsugar structures, biochemical characterization of enzymes in the pathway, and an understanding of genetic regulatory networks governing pathway expression will facilitate efforts to improve the endogenous defenses of Solanaceae crops with a variety of techniques ranging from marker-assisted selection to CRISPR/Cas9-mediated gene editing and synthetic biology approaches.

## 4. Materials and Methods 

### 4.1. Plant Material

Seeds of all *S. pennellii* accessions were obtained from the C.M. Rick Tomato Genetics Resource Center (TGRC; University of California, Davis, CA, USA). Seeds were treated with 2.6% sodium hypochlorite for 30 min and subjected to three 5-min rinses in de-ionized water before sowing on moist Whatman grade 1 filter paper (Sigma-Aldrich, St. Louis, MO, USA) in Petri dishes. Seeds were kept in a dark at room temperature and transplanted upon germination. Additional details of plant growth are shown in Appendix A.

### 4.2. Acylsugar Extraction

Single leaflets from the youngest fully expanded leaves of individual *S. pennellii* plants at 16 weeks post-germination were harvested and placed into pre-washed 10 mm × 75 mm borosilicate glass test tubes. Leaflets were collected from six individual plants of each *S. pennellii* accession, with an empty test tube included as a process blank. To each tube, 1 mL of a 3:3:2 mixture of acetonitrile/isopropanol/water containing 0.1% formic acid and 0.25 µM telmisartan internal standard was added. Tubes were vortexed for 30 s and solvent decanted into 2-mL glass autosampler vials. Equal volumes of each extract (excluding the process blank) were combined to create a pooled quality control (QC) sample. Vials were sealed with polytetrafluoroethylene (PTFE)-lined caps and stored at −20 °C for later processing.

### 4.3. Metabolomic Analysis by UHPLC–MS

Aliquots of *S. pennellii* acylsugar extracts, process blank, and QC sample were diluted 100-fold in 1:1 methanol/water containing 0.1% formic acid in new 2-mL autosampler vials. Five aliquots of the diluted process blank and QC samples were prepared and analyzed. Analyte samples were injected in a randomized order while process blank and QC samples were injected at regular intervals. Samples were subjected to UHPLC–MS analysis using an Acquity I-class pump coupled to a G2-XS QToF mass spectrometer (Waters Corporation, Milford, MA, USA). Separations were performed by reverse phase (C18) chromatography using a 20 min gradient of ammonium formate (pH 2.8) and acetonitrile. Analytes were ionized using positive-mode electrospray ionization and high-resolution mass spectra were acquired in continuum format from 2 to 18 min using quasi-simultaneous acquisition of low- and high-energy spectra (MS^E^). The UHPLC–MS method is detailed in its entirety in Appendix A.

### 4.4. Untargeted Metabolomics Data Processing

For untargeted metabolomic analysis, data were initially processed using Progenesis QI v2.4 software (Nonlinear Dynamics Ltd., Newcastle, UK). Leucine enkephalin lockmass correction (*m/z* 556.2766) was applied during run importation and all runs were aligned to retention times of a bulk pool run automatically selected by the software. Peak picking was carried out on features eluting between 2.15 and 14.5 min using an automatic sensitivity level of five (most sensitive) without restrictions on minimum chromatographic peak width. This resulted in the detection of 2361 compound ions. Spectral deconvolution was carried out, considering the following possible adduct ions: [M+H-H_2_O]^+^, [M+H], [M+NH_4_]^+^, [M+Na]^+^, [M+K]^+^, [M+C_2_H_8_N]^+^, [2M+H]^+^, [2M+NH_4_]^+^, [2M+Na]^+^, [2M+K]^+^, [2M+C_2_H_8_N]^+^. After deconvolution, 1559 compound ions remained.

To remove features from the dataset introduced by solvents, glassware, or instrumentation, several filters were applied to the 1559 compound ions remaining after deconvolution. Compounds with the highest mean abundance in process blank samples, maximum abundance less than 0.5% of the most abundant compound in the dataset, or a coefficient of variation >20% across QC samples were excluded from the dataset. This reduced the total number of metabolic features to 54.

Further analysis of compound signals extracted by Progenesis QI software was executed using EZinfo v3.0.2 software (Umetrics, Umeå, Sweden). For principal component analysis (PCA) and orthogonal partial least squares/projection to latent structures discriminant analysis (OPLS–DA), data were subjected to Pareto scaling. Generation of OPLS–DA models was carried out as follows: for each model, the relevant data files were divided into three subsets, each subset containing data files representing two of six biological replicates from each accession considered by the model. The first data subset contained data files representing the first pair of biological replicates from each relevant accession in the randomized injection list, while the second and third subsets contained the second and third pairs of biological replicates, respectively. The three data subsets, each representing one third of the relevant data, were used as training sets to generate three independent OPLS–DA models. Each model was then used to classify the remaining two thirds of the data not used in generation of the model, representing four of six biological replicates from each accession considered by the model. All OPLS–DA model statistics reported represent averages of the three independent models.

For all metabolic features extracted with Progenesis QI and used in downstream analyses with EZinfo, spectra were interpreted using MassLynx v4.2 software (Waters Corporation, Milford, MA, USA). Accurate masses of all features in all raw data files were obtained by applying the Continuous Lockmass Correction feature of the Accurate Mass Measure module. All precursor ions (annotated as either [M+NH_4_]^+^ or [M+H]^+^ adducts) were selected from the low-energy function while all fragment ions were assigned based on the high-energy function. Observed *m/z* values for precursor and product ions as well as neutral loss masses were compared to theoretical values generated using ChemDraw v19.0 software (PerkinElmer, Inc., Waltham, MA, USA). For acylsugars, molecular formulas were determined by comparing accurate *m/z* values of [M+NH_4_]^+^ pseudomolecular ions to theoretical *m/z* values of hypothetical acylsugar [M+NH_4_]^+^ adducts. The molecular formulas of all acyl chain components from individual acylsugars were inferred by a similar process using ketene and fatty acid neutral losses from pseudomolecular precursor ions observed in the high-energy function. While acylium product ions representing acyl chains appear in many spectra, their occurrence is inconsistent across compounds, especially in those of low abundance. Therefore, all acyl chain assignments were made using the neutral loss data, which could be unambiguously interpreted for all spectra. For flavonoids, molecular formulas were determined by comparing accurate *m/z* values of [M+H]^+^ pseudomolecular ions to theoretical *m/z* values of hypothetical flavonoid [M+H]^+^ adducts.

### 4.5. Acylsugar Quantification

Acylsugars were quantified from untargeted UHPLC–MS data by integration of extracted ion chromatogram peaks using the QuanLynx module of MassLynx software (Waters Corporation). All acylsucroses and acylglucoses detected in the metabolomics dataset were quantified using a standard curve of two each of purified acylsucroses and acylglucoses [S3:12(4,4,4), S3:18(4,4,10)-1, G3:12(4,4,4), and G3:18(4,4,10)-1] of authenticated concentrations at 0.3125, 0.625, 1.25, 2.5, and 5.0 µM. Acylsugars containing fewer than 18 carbons in all acyl chains were quantified using the G3:12(4,4,4) or S3:12(4,4,4) response factor while acylsugars containing 18 or more carbons were quantified using the response factor of G3:18(4,4,10)-1 or S3:18(4,4,10)-1. All quantifications were performed using extracted ion chromatograms of the *m/z* value for the relevant M+NH_4_^+^ adduct using a mass window of *m/z* 0.05. When multiple acylsugar isomers (including anomers) were present, all acylsugars of a given molecular formula were quantified using a single extracted ion chromatogram. The retention time window was adjusted for each compound based on the number of isomers and retention time differences between isomers. Telmisartan was used as an internal reference for all quantifications.

For the quantification of total acylsugar cores (i.e., sucrose and glucose), acylsugar extracts or purified acylsugars were saponified and the sugar core quantified using UHPLC–MS-MS. For each acylsugar analyte or standard, a 20-µL aliquot was evaporated to dryness in a 1.7-mL microfuge tube using a vacuum centrifuge and dissolved in 200 µL of a 1:1 methanol/3 M aqueous ammonia solution. The saponification reactions were incubated at room temperature for 48 h, at which point solvent was removed by vacuum centrifuge at room temperature. The dried residue was dissolved in 200 µL of 10 mM ammonium bicarbonate (pH 8.0) in 90% acetonitrile containing 0.5 μM ^13^C_12_-sucrose and 0.5 μM ^13^C_6_-glucose as internal standards and transferred to a 2-mL glass vial. Samples were subjected to UHPLC–MS-MS analysis. Levels of sucrose and glucose were quantified using a standard curve of the corresponding sugar at final concentrations of 3.13, 6.25, 12.5, 25, and 50 µM. Details of the UHPLC–MS-MS method are shown in Appendix A. 

### 4.6. RNA Extraction, cDNA Synthesis, and qPCR

Relative *ASFF1* transcript levels were measured using a published method [22]. Briefly, single leaflets from the youngest fully-expanded leaf of three biological replicates each of 12-week-old *S. pennellii* LA0716 and LA2963 plants were harvested and powdered under liquid nitrogen prior to RNA extraction using the RNeasy Plant Mini Kit (Qiagen, Hilden, Germany) and cDNA synthesis using SuperScript III reverse transcriptase (Invitrogen, Carlsbad, CA, USA). The gDNA_EF-1a_F/R primers were used to confirm the absence of gDNA contamination in synthesized cDNA by PCR. qRT-PCR analysis was carried out using SYBR Green PCR Master Mix on a QuantStudio 7 Flex Real-Time PCR System (Applied Biosystems, Warrington, UK). RT_ASFF_F and RT_ASFF_R primers were used to detect the *ASFF1* transcript; RT_EF-1a_F/R, RT_actin_F/R, and RT_ ubiquitin_F/R primers were used to detect transcripts of the *EF-1*α, *actin*, and *ubiquitin* genes, respectively (Appendix A). The cycling conditions were as follows: 48 °C for 30 min, 95 °C for 10 min, 40 cycles of 95 °C for 15 s and 60 °C for 1 min. Relative levels of *ASFF1* transcript were determined using the ΔΔCt method [49] and normalized to the geometric mean of *EF-1α*, *actin*, and *ubiquitin* transcript levels.

### 4.7. Acylsugar Purification

Purifications were performed using a Waters 2795 Separations Module (Waters Corporation) and an Acclaim 120 C18 HPLC column (4.6 mm × 150 mm, 5 μm; ThermoFisher Scientific, Waltham, MA, USA) with a column oven temperature of 30 °C and flow rate of 2 mL/min. For acylsucrose purification, the mobile phase consisted of water (solvent A) and acetonitrile (solvent B). For acylglucose purification, methanol was used as solvent B. Fractions were collected using a 2211 Superrac fraction collector (LKB Bromma, Stockholm, Sweden).

For purification of acylsucroses, acylsugars were extracted from mature plants of the *S. pennellii* LA0716 *ASFF1-1* mutant [22], which exclusively accumulates acylsucroses. Surface metabolites from ~75 g leaflets were extracted in 500 mL methanol containing 0.1% formic acid. This extract was dried under vacuum with a rotary evaporator and the resulting residue dissolved in ~3 mL acetonitrile containing 0.1% formic acid. Quantification of this solution by UPLC–MS-MS indicated a concentration of ~150 mM total acylsucroses. This extract was diluted 14-fold in 70% acetonitrile containing 0.1% formic acid. Acylsucroses were purified by pooling fractions from 10 injections of 50 µL each. Linear gradients of 45% B at 0 min, 60% B at 30 min, 100% B at 30.01 min held until 35 min, and 45% B at 35.01 min held until 40 min, were used. Fractions were collected at 10-s intervals into tubes containing 300 µL of 0.1% formic acid in water. The S3:12(4,4,4) compound eluted between 1 and 2 min; the S3:18(4,4,10)-1 compound eluted between 12 and 14 min; the S3:18(4,4,10)-2 compound eluted between 14 and 16 min; the S3:19(4,5,10)-1 compound eluted between 17 and 19 min; and the S3:19(4,5,10)-2 compound eluted between 19 and 21 min.

For the purification of acylglucoses, acylsugars were extracted from mature plants of *S. pennellii* LA0716, which accumulates > 90% acylglucoses [26]. Surface metabolites were extracted from ~75 g leaflets as described for acylsucrose purification. UHPLC–MS-MS quantification of the resulting solution indicated ~500 mM acylglucose concentration. This extract was diluted 20-fold in 1:1 methanol/water containing 0.1% formic acid. Acylglucoses were purified by pooling fractions after 20 injections of 50 µL each. Linear gradients of 5% B at 0–1 min, 60% B at 2 min, 100% B at 32 min held until 35 min, and 5% B at 36 min held until 40 min, were used. The G3:12(4,4,4) compound eluted between 6 and 7 min; the G3:18(4,4,10)-1 compound eluted between 17 and 18 min; the G3:18(4,4,10)-2 compound eluted between 18 and 19 min; the G3:19(4,5,10)-1 compound eluted between 20 and 21 min; and the G3:19(4,5,10)-2 compound eluted between 21 and 22 min.

The purity of acylsugar fractions was verified by UHPLC–MS using an LC-20AD HPLC (Shimadzu, Kyoto, Japan) coupled to a G2-XS QToF mass spectrometer (Waters Corporation, Milford, MA, USA). Separations were performed using an Ascentis Express C18 HPLC column (2.1 mm × 100 mm, 2.7 µm; Supelco, Bellefonte, PA). The mobile phases consisted of 100 mM ammonium formate, pH 3.4 (solvent A) and 100 mM ammonium formate, pH 3.4, in 90% methanol (solvent B). Five-microliter aliquots were injected onto the column and eluted with linear gradients of 5% B at 0-1 min, 60% B at 1.01 min, 100% B at 8 min, and 5% B at 8.01–10 min. The solvent flow rate was 0.4 mL/min and the column temperature was 40 °C. Analyses were performed using positive-ion mode electrospray ionization and sensitivity mode analyzer parameters. The source parameters were: capillary voltage at 3.00 kV, sampling cone voltage at 40 V, source offset at 80 V, source temperature at 100 °C, desolvation temperature at 350 °C, cone gas flow at 50.0 L/hour, and desolvation gas flow at 600.0 L/hour. Quasi-simultaneous mass spectrum acquisition at low and high collision energy conditions (MS^E^) was performed over an *m/z* range of 50 to 1500 with a scan time of 0.5 s. Adduct ions were obtained using a collision potential of 6.0 V; fragment ions were obtained using a collision potential ramp of 15 to 40 V. Spectra were acquired in centroid format.

Pure acylsugar fractions were pooled and solvent removed using a vacuum centrifuge. Residues were reconstituted in 1 mL 3:3:2 acetonitrile/isopropanol/water with 0.1% formic acid, transferred to 2-mL glass autosampler vials, sealed with PTFE-lined caps, and stored at −20 °C. Aliquots of purified acylsugars were quantified using the saponification method described above.

### 4.8. NMR Spectroscopy

NMR spectra (^1^H, gCOSY, gHSQC, gHMBC, and ^1^H-^1^H *J*-resolved spectra) were collected at the Max T. Rogers NMR Facility at Michigan State University using a DDR 500 MHz NMR spectrometer (Agilent, Santa Clara, CA, USA) equipped with a 7600AS 96-sample autosampler running VnmrJ v3.2A software. ^13^C spectra were collected on the same instrument at 125 MHz. All spectra were referenced to non-deuterated chloroform solvent signals (δH = 7.26 (s) and δC = 77.2 (t) ppm). Additional details of the NMR data collection methods are shown in Appendix A. 

## Figures and Tables

**Figure 1 metabolites-10-00401-f001:**
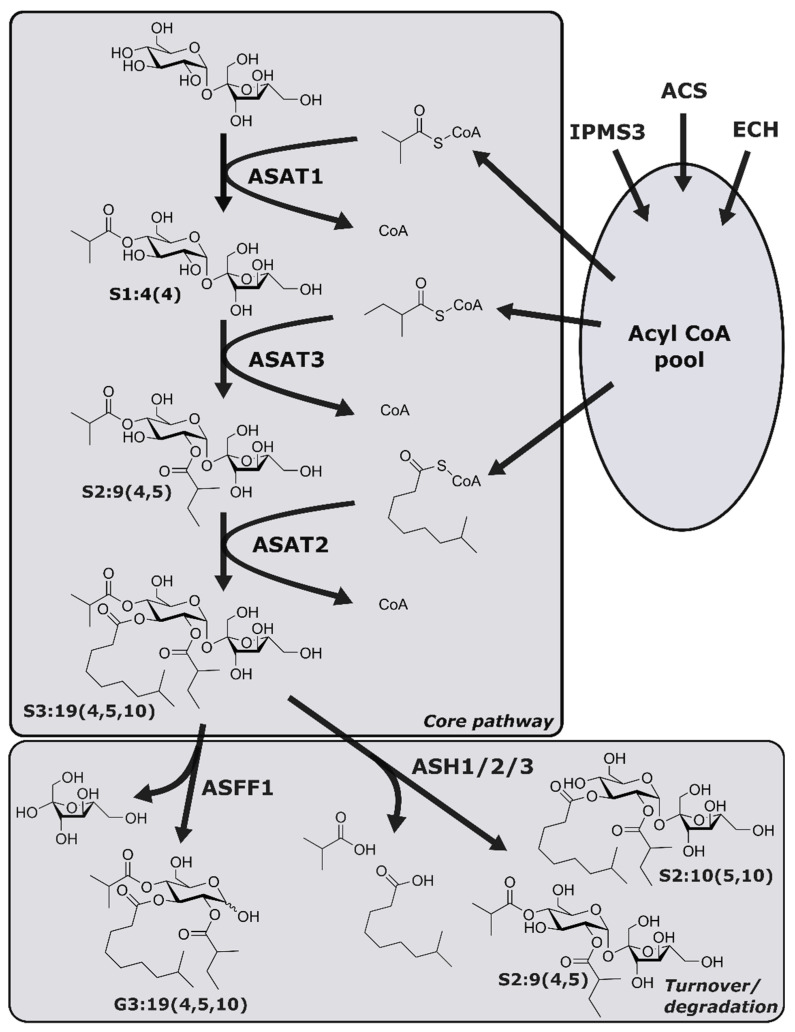
The acylsugar biosynthetic pathway in *S. pennellii*. Enzymes including acyl CoA synthetase (ACS), enoyl CoA hydratase (ECH), and isopropylmalate synthase 3 (IPMS3) contribute to production of the acyl CoA pool. Acylsugar acyltransferases (ASATs) constitute the core acylsugar pathway and transfer acyl chains from acyl CoA molecules to a sucrose core. ASFF1 and ASHs catalyze acylsugar turnover or degradation by hydrolyzing the fructose moiety of the sugar core and acyl chains, respectively. Acylsugar nomenclature is as follows: the first letter indicates the sugar core (“S” for sucrose, “G” for glucose); the number before the colon indicates the number of acyl chains; the number after the colon indicates the sum of carbons in all acyl chains; the numbers in parentheses indicate the number of carbons in individual acyl chains. ACS—acyl CoA synthetase; ASAT—acylsucrose acyltransferase; ASFF—acylsucrose fructofuranosidase; ASH—acylsugar hydrolase; CoA—coenzyme A; ECH—enoyl CoA hydratase; IPMS—isopropyl malate synthase.

**Figure 2 metabolites-10-00401-f002:**
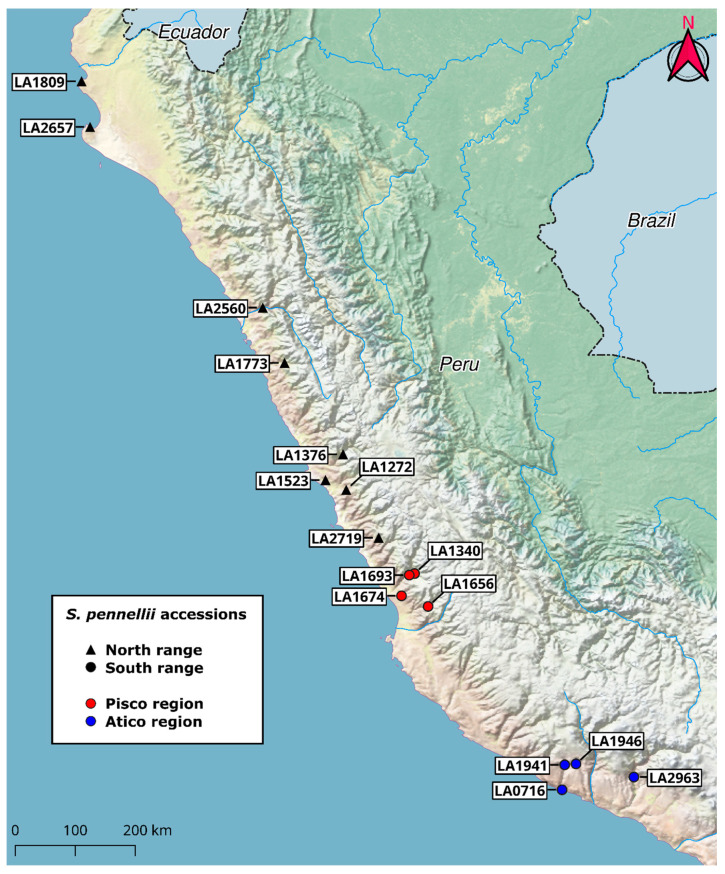
Locations of *S. pennellii* accessions used in this study across the geographic range of the species in Peru. Accessions classified as belonging to the north range are denoted with black triangles, those classified as belonging to the south range with circles. South range accessions are further classified by region (red for Pisco, blue for Atico). Global positioning system (GPS) coordinates for accession locations were provided by the C.M. Rick Tomato Genetics Resource Center.

**Figure 3 metabolites-10-00401-f003:**
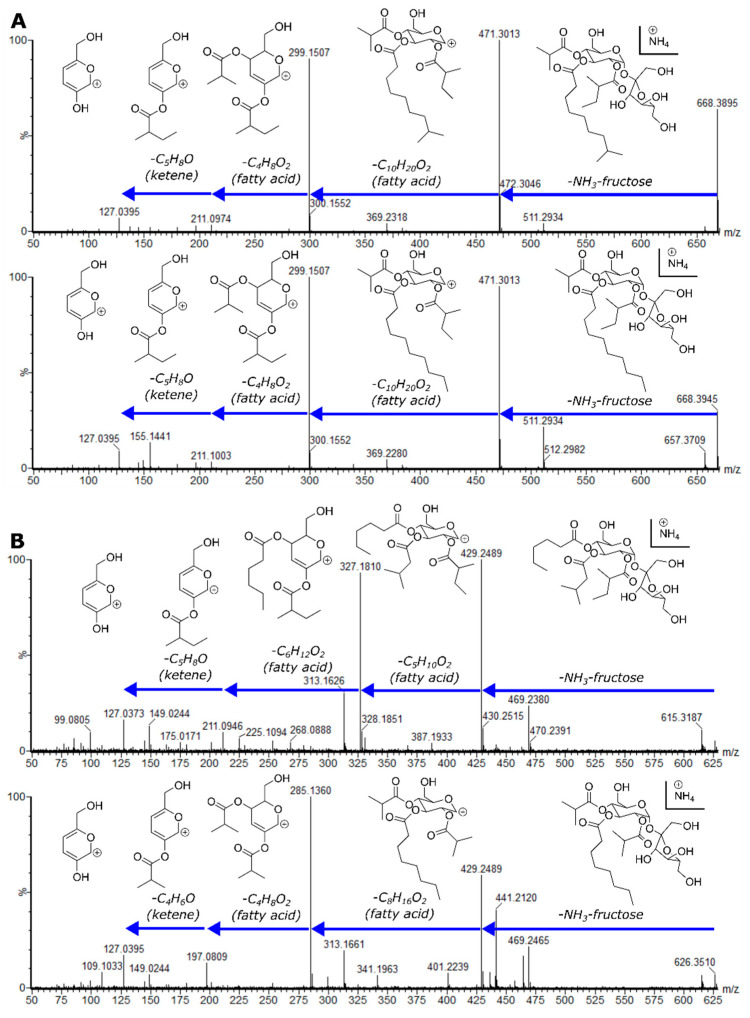
CID mass spectra of acylsugar structural isomers in positive-ion mode. (**A**) Mass spectra of S3:19(4,5,10)-1 (top) and S3:19(4,5,10)-2 (bottom). Structures of both compounds were resolved by NMR (Figure 4). (**B**) Mass spectra of S3:16(5,5,6) (top) and S3:16(4,4,8) (bottom). Structures of these compounds were not resolved by NMR, and the specific branching patterns and positions of acyl chains are unknown.

**Figure 4 metabolites-10-00401-f004:**
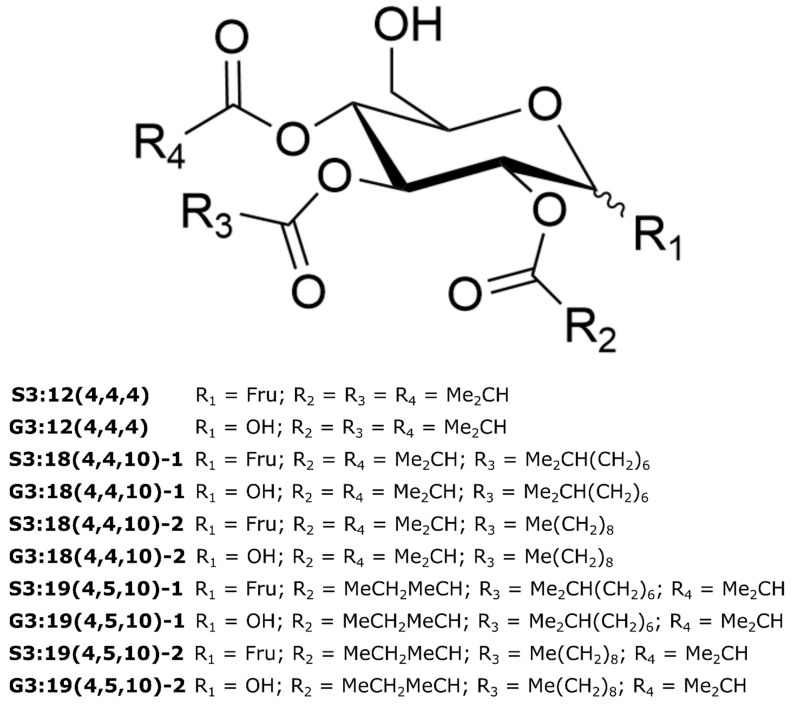
NMR-resolved structures of acylsugars purified from *S. pennellii*. For acylsucroses, the R_1_ group is observed only in the α configuration. For each acylglucose, two distinct anomers are observed with group R_1_ in either the α or β configuration. Fru = β-fructofuranose.

**Figure 5 metabolites-10-00401-f005:**
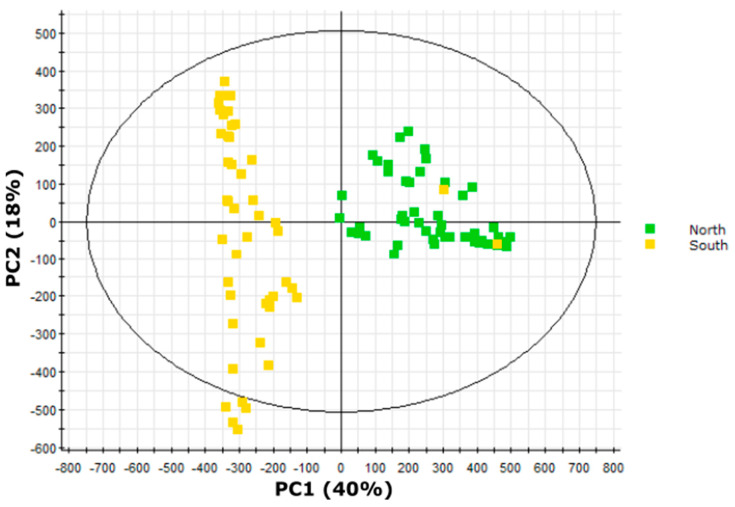
Principal component analysis (PCA) scores plot of samples from 16 *S. pennellii* accessions from across Peru separated by abundances of 54 metabolite features identified in trichome extracts by UHPLC–HR-MS. Samples from the North range are indicated in green, while samples from the South range are indicated in yellow (see Figure 2 for details on geographic range). Principal component 1 (PC1) accounted for approximately 40% of the variance in the dataset and drove strong separation between north and south accessions, while PC2 accounted for 18% of the variance and associated primarily with variation within the south range accessions.

**Figure 6 metabolites-10-00401-f006:**
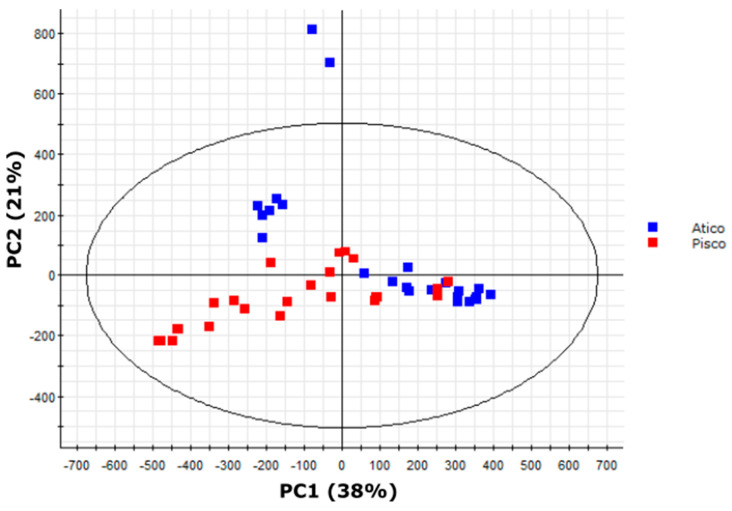
PCA scores plot of samples from eight *S. pennellii* accessions from the southern portion of the range of the species in Peru separated by abundances of 54 metabolite features identified in trichome extracts by UHPLC–HR-MS. Samples from the Atico region are indicated in blue, while samples from the Pisco region are indicated in red (see Figure 2 for details on regions). Separation is observed between Atico and Pisco samples. However, Atico region samples exhibit bimodal clustering. PC1 accounted for 38% of variance and described most of the variation between accession LA2963 samples and other Atico region accessions, while PC2 accounted for 21% of variance and described primarily variation within the main Atico cluster.

**Figure 7 metabolites-10-00401-f007:**
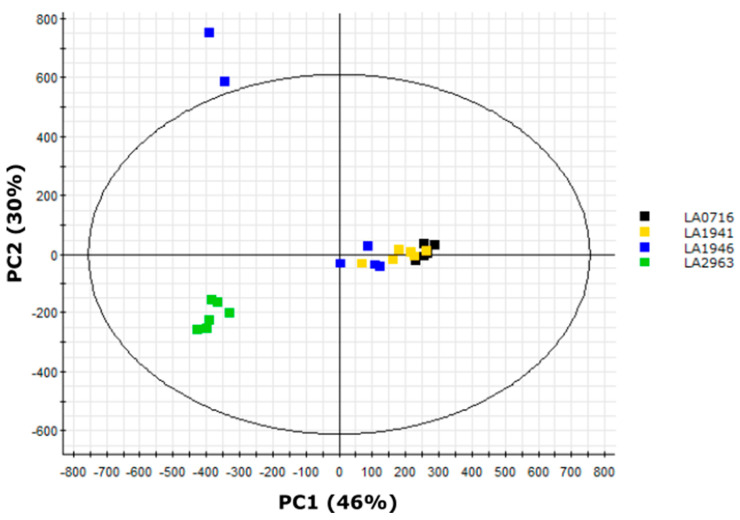
PCA scores plot of samples from four *S. pennellii* accessions in the Atico region of Peru separated by abundances of 54 metabolite features identified in trichome extracts by UHPLC–HR-MS. Samples from accession LA0716 are indicated in black, samples from accession LA1941 in yellow, samples from accession LA1946 in blue, and samples from accession LA2963 in green (see Figure 2 for details of the Atico region). PC1 accounted for 46% of variance and described most of the variation between accession LA2963 samples and other Atico region accessions, while PC2 accounted for 30% of variance and described primarily variation within the main Atico cluster.

**Figure 8 metabolites-10-00401-f008:**
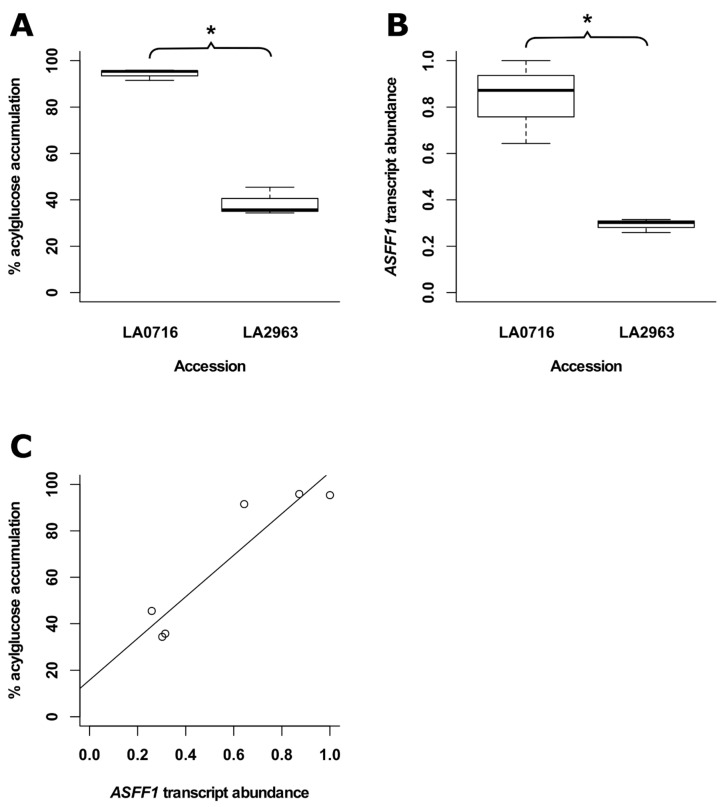
Analysis of acylglucose accumulation and *ASFF1* transcript abundance in paired leaflets of *S. pennellii* accessions LA0716 and LA2963. (**A**) Percentage of total acylsugars accumulating as acylglucoses. (**B**) Relative abundance of *ASFF1* transcripts. (**C**) Linear regression of *ASFF1* transcript abundance and percentage of percentage of acylsugars accumulating as acylglucoses (data points represented as open circles; R^2^ = 0.84). “*” indicates *p* < 0.05 (analysis of variance); *n* = 3 for both accessions.

**Table 1 metabolites-10-00401-t001:** Annotations of acylsugars in *S. pennellii*. Acylsugar nomenclature is as follows: the first letter indicates the sugar core (“S” for sucrose, “G” for glucose); the number before the colon indicates the number of acyl chains; the number after the colon indicates the sum of carbons in all acyl chains; the numbers in parentheses indicate the number of carbons in individual acyl chains. MSI = Metabolomics Standards Initiative score of confidence in annotation [41]. RT = retention time (min); *m/z*_acc_ = accurate [M+NH_4_]^+^ mass measured; *m/z*_ex_ = exact mass calculated from formula; Δm (ppm) = parts per million error between *m/z*_ex_ and *m/z*_acc_; fragment *m/z* = ions used for acyl chain determinations.

Name	MSI	RT	Formula	*m/z* _acc_	*m/z* _ex_	Δm (ppm)	Fragment *m/z*
Triacylsucroses							
S3:12(4,4,4)	1	2.21	C_24_H_40_O_14_	570.2778	570.2756	3.9	373.1872, 285.1326, 197.0809, 127.0395
S3:13(4,4,5)	3	2.34	C_25_H_42_O_14_	584.2926	584.2913	2.2	387.2010, 299.1507, 197.0809, 127.0395
S3:14(4,5,5)	3	2.59	C_26_H_44_O_14_	598.3075	598.3069	1.0	401.2178, 313.1668, 211.0951, 127.0396
S3:15(5,5,5)	3	3.00	C_27_H_46_O_14_	612.3229	612.3226	0.5	415.2348, 313.1661, 211.0974, 127.0395
S3:16(5,5,6)	3	3.44	C_28_H_48_O_14_	626.3392	626.3382	1.6	429.2489, 327.1810, 211.0946, 127.0373
S3:16(4,4,8)	3	3.67	C_28_H_48_O_14_	626.3387	626.3382	0.8	429.2489, 285.1360, 197.0809, 127.0395
S3:17(4,5,8)	3	4.23	C_29_H_50_O_14_	640.3543	640.3539	0.6	443.2710, 299.1541, 211.0974, 127.0395
S3:17(4,4,9)	3	4.43	C_29_H_50_O_14_	640.3536	640.3539	−0.5	443.2646, 285.1341, 197.0773, 127.0396
S3:18(4,4,10)-1	1	5.45	C_30_H_52_O_14_	654.3699	654.3695	0.6	457.2864, 285.1360, 197.0837, 127.0395
S3:18(4,4,10)-2	1	5.71	C_30_H_52_O_14_	654.3699	654.3695	0.6	457.2864, 285.1360, 197.0837, 127.0395
S3:19(4,5,10)-1	1	6.24	C_31_H_54_O_14_	668.3856	668.3852	0.6	471.3013, 299.1507, 211.0974, 127.0395
S3:19(4,5,10)-2	1	6.52	C_31_H_54_O_14_	668.3855	668.3852	0.5	471.3013, 299.1507, 211.1003, 127.0395
S3:20(5,5,10)	3	7.57	C_32_H_56_O_14_	682.4011	682.4008	0.4	485.3103, 313.1661, 211.0974, 127.0395
S3:20(4,4,12)	3	8.20	C_32_H_56_O_14_	682.4009	682.4008	0.2	485.3146, 285.1360, 197.0837, 127.0395
S3:21(5,5,11)	3	8.56	C_33_H_58_O_14_	696.4166	696.4165	0.1	499.3259, 313.1661, 211.0974, 127.0395
S3:21(4,5,12)	3	9.14	C_33_H_58_O_14_	696.4161	696.4165	−0.6	499.3259, 299.1472, 211.0974, 127.0395
S3:22(5,5,12)	3	10.26	C_34_H_60_O_14_	710.4319	710.4321	−0.3	513.3442, 313.1661, 211.0974, 127.0395
Triacylsucroses (cont’d)							
S3:23(5,6,12)	3	11.24	C_35_H_62_O_14_	724.4471	724.4478	−1.0	527.3616, 327.1810, 211.0946, 127.0373
Triacylglucoses							
G3:12(4,4,4)	1	2.76; 2.84	C_18_H_30_O_9_	408.2235	408.2228	1.7	373.1872, 285.1326, 197.0809, 127.0395
G3:13(4,4,5)	3	3.12; 3.24	C_19_H_32_O_9_	422.2392	422.2385	1.7	387.2014, 299.1501, 197.0801, 127.0396
G3:14(4,5,5)	3	3.70; 3.83	C_20_H_34_O_9_	436.2547	436.2541	1.4	401.2178, 299.1466, 211.0951, 127.0374
G3:15(5,5,5)	3	4.42; 4.58	C_21_H_36_O_9_	450.2705	450.2698	1.6	415.2308, 313.1626, 211.0974, 127.0395
G3:16(5,5,6)	3	5.23; 5.41	C_22_H_38_O_9_	464.2859	464.2854	1.1	429.2489, 327.1810, 211.0946, 127.0395
G3:16(4,4,8)-1	3	5.56; 5.80	C_22_H_38_O_9_	464.2861	464.2854	1.5	429.2529, 285.1360, 197.0809, 127.0395
G3:16(4,4,8)-2	3	5.80; 6.04	C_22_H_38_O_9_	464.2857	464.2854	0.7	429.2529, 285.1360, 197.0809, 127.0395
G3:17(4,5,8)-1	3	6.46; 6.72	C_23_H_40_O_9_	478.3011	478.3011	0.0	443.2628, 299.1472, 211.0946, 127.0395
G3:17(4,5,8)-2	3	6.71; 6.99	C_23_H_40_O_9_	478.3008	478.3011	−0.6	443.2628, 299.1472, 197.0781, 127.0373
G3:18(4,4,10)-1	1	8.03; 8.33	C_24_H_42_O_9_	492.3168	492.3167	0.2	457.2779, 285.1326, 197.0809, 127.0395
G3:18(4,4,10)-2	1	8.33; 8.64	C_24_H_42_O_9_	492.3170	492.3167	0.6	457.2779, 285.1326, 197.0809, 127.0395
G3:19(4,5,10)-1	1	9.05; 9.34	C_25_H_44_O_9_	506.3328	506.3324	0.8	471.2970, 299.1472, 211.0974, 127.0395
G3:19(4,5,10)-2	1	9.34; 9.66	C_25_H_44_O_9_	506.3328	506.3324	0.8	471.2970, 299.1507, 211.0946, 127.0395
G3:20(5,5,10)	3	10.47; 10.72	C_26_H_46_O_9_	520.3486	520.3480	1.2	485.3146, 313.1661, 211.0974, 127.0395
G3:20(4,4,12)	3	11.10; 11.42	C_26_H_46_O_9_	520.3483	520.3480	0.6	485.3103, 285.1326, 197.0809, 127.0395
G3:21(5,5,11)	3	11.47; 11.75	C_27_H_48_O_9_	534.3637	534.3637	0.0	499.3215, 313.1626, 211.0974, 127.0373
G3:21(4,5,12)	3	12.10; 12.40	C_27_H_48_O_9_	534.3636	534.3637	−0.2	499.3290, 299.1507, 211.0974, 127.0395
Triacylglucoses (cont’d)							
G3:22(5,5,12)	3	13.10; 13.36	C_28_H_50_O_9_	548.3794	548.3793	0.2	513.3442, 313.1661, 211.0974, 127.0395
G3:23(5,6,12)	3	14.02; 14.29	C_29_H_52_O_9_	562.3938	562.3950	−2.1	527.3471, 327.1848, 211.0960, 127.0372
Tetraacylglucoses							
G4:14(2,4,4,4)	3	3.54; 3.79	C_20_H_32_O_10_	450.2343	450.2334	2.0	415.1946, 327.1417, 239.0891, 197.0809, 127.0373
G4:15(2,4,4,5)	3	4.10; 4.45	C_21_H_34_O_10_	464.2502	464.2491	2.4	429.2162, 341.1562, 239.0922, 197.0837, 127.0395

**Table 2 metabolites-10-00401-t002:** Annotations of flavonoids in *S. pennellii*. MSI = Metabolomics Standards Initiative score of confidence in annotation [41]. RT = retention time (min); *m/z*_acc_ = accurate [M+H]+ mass measured; *m/z*_ex_ = exact mass calculated from formula; Δm (ppm) = parts per million error between *m/z*_ex_ and *m/z*_acc_; core = putative flavonol core based on molecular formula; # Me = number of methyl groups based on molecular formula and mass spectrum (Appendix A).

Name	MSI	RT	Formula	*m/z* _acc_	*m/z* _ex_	Δm (ppm)	Core	# Me
Flavonoid A	3	3.04	C_17_H_14_O_6_	315.0869	315.0863	1.9	kaempferol	2
Flavonoid C	3	3.17	C_18_H_16_O_7_	345.0980	345.0969	3.2	quercetin	3
Flavonoid D	3	4.00	C_19_H_18_O_7_	359.1137	359.1125	3.3	quercetin	4
Flavonoid B	3	4.90	C_18_H_16_O_6_	329.1025	329.1020	1.5	kaempferol	3

**Table 3 metabolites-10-00401-t003:** Acylsugar accumulation and percent acylglucose in accessions of *S. pennellii*, as determined by UHPLC–MS-MS. Values are presented as mean ± SD (*n* = 6). Results of analysis of variance and Tukey’s mean-separation test are indicated as letters. Accessions that do not have at least one letter in common are significantly different from one another (*p* < 0.05). The range and region of each accession within Peru are also indicated.

Accession	Total Acylsugars (µmol/g DW)	Tukey’s MST	% Acylglucose	Tukey’s MST	Range	Region
LA1809	136 ± 27	B	69 ± 4	CDE	North	
LA2657	133 ± 28	B	56 ± 6	EF	North	
LA2560	340 ± 64	A	65 ± 6	CDE	North	
LA1773	237 ± 98	AB	66 ± 2	CDE	North	
LA1376	261 ± 105	AB	70 ± 7	CDE	North	
LA1523	158 ± 68	B	65 ± 8	CDE	North	
LA1272	163 ± 109	B	58 ± 4	DEF	North	
LA2719	218 ± 44	AB	70 ± 3	BCDE	North	
LA1340	166 ± 39	B	80 ± 10	ABC	South	Pisco
LA1693	193 ± 88	AB	77 ± 7	ABCD	South	Pisco
LA1674	248 ± 83	AB	90 ± 4	A	South	Pisco
LA1656	269 ± 54	AB	90 ± 4	AB	South	Pisco
LA1946	257 ± 94	AB	82 ± 20	ABC	South	Atico
LA1941	244 ± 66	AB	95 ± 2	A	South	Atico
LA2963	183 ± 38	B	42 ± 6	F	South	Atico
LA0716	238 ± 75	AB	95 ± 2	A	South	Atico

**Table 4 metabolites-10-00401-t004:** Orthogonal partial least squares/projection to latent structures discriminant analysis (OPLS–DA) model performance. The table indicates the percentage of test samples that each model classified correctly, incorrectly, or was unable to classify.

	*n*	% Correct	% Incorrect	% Unknown
Full range (N = 96)				
North	48	100	0	0
South	48	94	4	2
South range (N = 48)				
Pisco	24	67	4	29
Atico	24	77	4	19
Atico region (N = 24)				
LA0716/LA1941/LA1946	18	97	0	3
LA2963	6	100	0	0

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
