# Peer review of "An Integrated Analytical Approach Reveals Trichome Acylsugar Metabolite Diversity in the Wild Tomato Solanum pennellii"

_metabolites, 2020, doi:10.3390/metabo10100401_

Round 1

Reviewer 1 Report

The authors performed an untargeted metabolomic analysis by LC-MS to study the acylsugar metabolite diversity present in the trichomes of the wild tomato Solanum pennellii. They used complementary analytical techniques and multivariate statistics to make an important contribution in the trichome chemistry of S. pennelli. I found the manuscript particularly interesting showing rather comprehensive results including a variety of analytical techniques and methods, such as LC-MS-based dereplication, quantification, Rt-qPCR, isolation and NMR. The originality, significance and quality of the content is high and deserves to be published. However, some important information is lacking in the methodology (see below) and I have some concerns about the methods used in the multivariate statistics and qPCR that require a major revision.

Please find below my comments related to each section:

Results:

  1. For the ease of interpretation, I suggest including an additional image with the chemical structures, names, and codes of the acylsugars and flavonoids reported in Tables 1 and 2, as the current the codes do not provide much information to the reader.
  2. Please consider following the Metabolomics Standards Initiative regarding the confidence level achieved in the annotation of metabolites. These levels (1-4) can be included as a separate column in tables 1 and 2, so that the readers can assess how confident were the authors in assigning a specific chemical identity.
  3. If there are rather quantitative differences in the accumulation of acylglucoses between southern and northern accessions (as demonstrated by the quantification results in Table 3), it does not make sense to me to apply the UV scaling in the PCA, as the main disadvantage of this method is the inflation of measurement errors or low concentrated metabolites. As there are clear quantitative differences between northern and southern populations, I suggest using a scaling method that accounts for those differences such as Pareto or center scaling. Furthermore, the same scaling method should be applied for both unsupervised (PCA) and supervised tests (OPLS-DA) to be comparable (see comment below).
  4. The authors reported that five metabolites have the strongest quantitative correlation with either sample class (lines 271 to 276). However, it was not clear the method followed to identify those metabolites. Were they pointed by analysis of the OPLS-DA S-plot, loadings plot, etc?. If so, including an image (as supplementary) highlighting the discriminant metabolites of each group would be helpful. Further comments on whether these five metabolites confirm the previous quantifications results in Table 3 can be worth mentioning.

Methods:

  1. Please consider making the raw chromatographic data publicly available in metabolomic repositories such as MetaboLights, MASSIVE, etc.
  2. Basic details necessary for reproducing the LC-MS experiments must be included in the main paper. For example, in the section 4.3, the authors did not specify the composition of the mobile phase and its gradient, as well as the flow rate. Furthermore, parameters used in the MS including the resolution, ionization mode, temperature, and capillary voltage were also missing. Although some of this information is included in the supplementary Table S2, key parameters must be also included in this initial section, similar to what they reported in section 4.7, which in turn can be summarized and make reference to the previous section 4.3.
  3. Authors should clarify why two different scaling methods (Unit variance and Pareto) were used for the PCA and OPLS-DA analysis. These two scaling methods emphasize different aspects of the dataset and have different limitations and advantages. Therefore, unless the same scaling method in used both in the PCA and OPLS-DA, results are not comparable. I suggest standardizing the scaling method to be used (depending on the type of data, biological question, etc) and apply the same method for both analyses.
  4. In the qPCR analyses, I suggest that authors clarify whether they checked if the extracted RNA samples were free from gDNA? In qPCR residual gDNA coextracted in the RNA protocol could interfere in the results given by cDNA. Therefore, it is highly advisable to check for this and correct it in case of positive results by using commercial kits (e.g. DNase I treatment). Furthermore, the temperature program used in the qPCR must also be specified to allow replication of results. I also suggest reporting the primer efficiency for ASFF1 and including the melt curves obtained during qPCR (as supplementary) to check the absence of secondary bands in amplification products.

Author Response

Reviewer 1:

Results:

  1. For the ease of interpretation, I suggest including an additional image with the chemical structures, names, and codes of the acylsugars and flavonoids reported in Tables 1 and 2, as the current the codes do not provide much information to the reader.

Figure 4 (page 13) indicates the structures for all compounds in the dataset for which the   structure is known (MSI class 1). However, we have included a third supplemental file   (supp3.docx) that contains structural information including IUPAC chemical names,   InChiKey and SMILES codes, and line drawings of unambiguous chemical structures for      all class 1 compounds.

  1. Please consider following the Metabolomics Standards Initiative regarding the confidence level achieved in the annotation of metabolites. These levels (1-4) can be included as a separate column in tables 1 and 2, so that the readers can assess how confident were the authors in assigning a specific chemical identity.

MSI annotation confidence classes are now contained in Tables 1 and 2 (pages 7-9; see lines 152, 163-4) and described in the text (lines 189-193).

  1. If there are rather quantitative differences in the accumulation of acylglucoses between southern and northern accessions (as demonstrated by the quantification results in Table 3), it does not make sense to me to apply the UV scaling in the PCA, as the main disadvantage of this method is the inflation of measurement errors or low concentrated metabolites. As there are clear quantitative differences between northern and southern populations, I suggest using a scaling method that accounts for those differences such as Pareto or center scaling. Furthermore, the same scaling method should be applied for both unsupervised (PCA) and supervised tests (OPLS-DA) to be comparable (see comment below).

We have re-analyzed the dataset using Pareto scaling for both PCA and OPLS-DA and prepared new figures accordingly. Figures 5-7 (pages 15, 16, and 18) have been replaced with appropriate alterations to figure legends. Lines 551-554 have been altered to reflect the new data processing methods.

  1. The authors reported that five metabolites have the strongest quantitative correlation with either sample class (lines 271 to 276). However, it was not clear the method followed to identify those metabolites. Were they pointed by analysis of the OPLS-DA S-plot, loadings plot, etc?. If so, including an image (as supplementary) highlighting the discriminant metabolites of each group would be helpful. Further comments on whether these five metabolites confirm the previous quantifications results in Table 3 can be worth mentioning.

The text has been altered to explain quantitative correlation (lines 285-289; 324-331; 362-369). Additionally, we have provided the requested S-plots as supplemental figures (Figs. S63-65). The manuscript already included reference to Table 3 as requested (lines 370-373).

Methods:

  1. Please consider making the raw chromatographic data publicly available in metabolomic repositories such as MetaboLights, MASSIVE, etc.

The raw data will be available via MetaboLights once submission is approved by the system administrators.

  1. Basic details necessary for reproducing the LC-MS experiments must be included in the main paper. For example, in the section 4.3, the authors did not specify the composition of the mobile phase and its gradient, as well as the flow rate. Furthermore, parameters used in the MS including the resolution, ionization mode, temperature, and capillary voltage were also missing. Although some of this information is included in the supplementary Table S2, key parameters must be also included in this initial section, similar to what they reported in section 4.7, which in turn can be summarized and make reference to the previous section 4.3.

We have added key details of the LC-MS method to the main text (lines 530-534). However, the method described in section 4.7 is distinct from that described in section 4.3 including use of different instrument, column, and solvent system. As such, we cannot reference section 4.3 in lieu of describing the method in section 4.7.

  1. Authors should clarify why two different scaling methods (Unit variance and Pareto) were used for the PCA and OPLS-DA analysis. These two scaling methods emphasize different aspects of the dataset and have different limitations and advantages. Therefore, unless the same scaling method in used both in the PCA and OPLS-DA, results are not comparable. I suggest standardizing the scaling method to be used (depending on the type of data, biological question, etc) and apply the same method for both analyses.

We have re-analyzed the dataset using Pareto scaling for both PCA and OPLS-DA and prepared new figures accordingly. Figures 5-7 (pages 15, 16, and 18) have been replaced with appropriate alterations to figure legends. Lines 551-554 have been altered to reflect the new data processing methods.

  1. In the qPCR analyses, I suggest that authors clarify whether they checked if the extracted RNA samples were free from gDNA? In qPCR residual gDNA coextracted in the RNA protocol could interfere in the results given by cDNA. Therefore, it is highly advisable to check for this and correct it in case of positive results by using commercial kits (e.g. DNase I treatment). Furthermore, the temperature program used in the qPCR must also be specified to allow replication of results. I also suggest reporting the primer efficiency for ASFF1 and including the melt curves obtained during qPCR (as supplementary) to check the absence of secondary bands in amplification products.

The RT-qPCR methods have been amended to incorporate the method used to check for gDNA contamination and the PCR cycling conditions (lines 612-621). Additionally, Table S4 has been amended to include primer efficiencies. Melt curves were not obtained and therefore are not included as supplemental figures.

Reviewer 2 Report

The present manuscript “An integrated analytical approach reveals trichome acylsugar metabolite diversity in the wild tomato Solanum pennellii” describes the acylsugar and flavonoid metabolic composition within trichomes of different accessions of Peruvian wild tomato Solanum pennellii. Authors have identified a characterized a variety of acylsucrose and acylglucose decorated with different acyl chains. Qualitative and quantitative differences in terms of acylsugar have been described based on geographical localization of the accessions studied. Moreover, contribution of the enzyme ASFF1 (acylsucrose fructofuranosidase1) to acylsugar pool composition improves the knowledge on the biosynthetic pathway.

Paper conclusions are based on solid experimental evidences although a connection between the phytochemical composition and physiological aspects would have contributed to better understand the current work.

Minor points

  1. Briefly describing their protective proprieties could help readers to contextualize the research.
  2. Are qualitative or quantitative differences in terms of fatty acid composition within northern and southern accession trichome?
  3. Lines 113-117: is trichome-deficient accession more susceptible to biotic stresses?
  4. Lines 162-177: text editing, indentation.
  5. Different accession have different composition, any biological interpretation related to host-pathogen interations.
  6. Lines 192-193: these intermediates have been already identified in Solanum trichomes? In vitro studies?
  7. Lines 213-216: criterial selection?
  8. Line 264-266: no possible to repeat the experiment? Probably it is necessary a better explication.
  9. Figure 6: total % of the variance less than 60%
  10. Flavonoids are included in the study but not in the title.
  11. References with first letter of every word in uppercase and others don't

Author Response

Reviewer 2:

  1. Briefly describing their protective proprieties could help readers to contextualize the research.

This is described in lines 42-46.

  1. Are qualitative or quantitative differences in terms of fatty acid composition within northern and southern accession trichome?

This is described in lines 63-66 and lines 299-304.

  1. Lines 113-117: is trichome-deficient accession more susceptible to biotic stresses?

To our knowledge, biotic stress tolerance of trichome-deficient S. pennellii accessions has not been reported in the literature.

  1. Lines 162-177: text editing, indentation.

This issue has been corrected.

  1. Different accession have different composition, any biological interpretation related to host-pathogen interations.

We did not conduct any host-pathogen studies, nor are we aware of published comprehensive studies addressing this in S. pennellii.

  1. Lines 192-193: these intermediates have been already identified in Solanum trichomes? In vitro studies?

This section has been clarified to indicate observation of mono- and di-acylated sucroses as in vitro intermediates (lines 205-206).

  1. Lines 213-216: criterial selection?

Reasons for compound selection are now described (lines 230-231).

  1. Line 264-266: no possible to repeat the experiment? Probably it is necessary a better explication.

Due to the scale of the experiment, we could not repeat it at the stage at which this discrepancy was identified. The reasons listed are our best hypotheses for the discrepancy.

  1. Figure 6: total % of the variance less than 60%

The variances seen in our data are typical considering the number of accessions compared. The 60% rule is typically applied in scenarios involving comparison of two treatments such a one wild-type group and one mutant group or a single treatment and single control group.

  1. Flavonoids are included in the study but not in the title.

We include a brief description of the observed flavonoids because they appear in the dataset. However, the analysis of metabolite diversity reported here centers on acylsugars and the underlying biochemistry of acylsugar biosynthesis.

  1. References with first letter of every word in uppercase and others don't

The references section has been amended to fix this discrepancy (lines 700-826).

Round 2

Reviewer 1 Report

The authors have corrected the weaknesses of the previous version and I would like to congratulate them on this interesting and scientifically rigorous work.